# Cellular Protein Trafficking: A New Player in Low-Temperature Response Pathway

**DOI:** 10.3390/plants11070933

**Published:** 2022-03-30

**Authors:** M. Arif Ashraf, Abidur Rahman

**Affiliations:** 1Biology Department, University of Massachusetts, Amherst, MA 01003, USA; arif.ashraf.opu@gmail.com; 2The United Graduate School of Agricultural Sciences, Iwate University, Morioka 020-8550, Japan; 3Department of Plant Biosciences, Faculty of Agriculture, Iwate University, Morioka 020-8550, Japan; 4Department of Plant Sciences, College of Agriculture and Bioresources, University of Saskatchewan, Saskatoon, SK S7N 5A8, Canada

**Keywords:** temperature stress, low temperature, CBF pathway, cell biology, climate change, protein trafficking

## Abstract

Unlike animals, plants are unable to escape unfavorable conditions, such as extremities of temperature. Among abiotic variables, the temperature is notableas it affects plants from the molecular to the organismal level. Because of global warming, understanding temperature effects on plants is salient today and should be focused not only on rising temperature but also greater variability in temperature that is now besetting the world’s natural and agricultural ecosystems. Among the temperature stresses, low-temperature stress is one of the major stresses that limits crop productivity worldwide. Over the years, although substantial progress has been made in understanding low-temperature response mechanisms in plants, the research is more focused on aerial parts of the plants rather than on the root or whole plant, and more efforts have been made in identifying and testing the major regulators of this pathway preferably in the model organism rather than in crop plants. For the low-temperature stress response mechanism, ICE-CBF regulatory pathway turned out to be the solely established pathway, and historically most of the low-temperature research is focused on this single pathway instead of exploring other alternative regulators. In this review, we tried to take an in-depth look at our current understanding of low temperature-mediated plant growth response mechanism and present the recent advancement in cell biological studies that have opened a new horizon for finding promising and potential alternative regulators of the cold stress response pathway.

## 1. Climate Change and Plant Growth

As sessile organisms, plants are unable to escape unfavorable conditions and can experience both biotic and abiotic stresses. Between these two, abiotic stress has become the most prominent cause of agricultural loss [1,2]. Damage caused by biotic stress can be solved by manipulating single target genes or receptors. However, abiotic stresses are intertwined, which increase the plants’ susceptibility and result in more damages [3]. For instance, the synergistic effect of drought and heat is more destructive compared to individual stress [3,4]. Exposure to low temperatures results in mechanical constraints, alters signaling molecules, and reduces osmotic pressure at the cellular level [5,6]. During submergence, plants experience a combination of flooding, salinity stress, and hypoxia at the same time [7,8]. Among abiotic variables, temperature is notable because it affects almost every molecule and reaction in the cell.

Recently, temperature fluctuations have become exceptionally common across the globe. According to the latest data from the National Centers for Environmental Information, USA (NOAA), the global land surface temperature for March 2018 was 1.5 °C (2.9 °F) above average and it was the seventh highest since global records began in 1880. At the same time, cooler-than-average conditions engulfed much of Europe and western Russia during March 2018. For example, Lyon, in France, observed an average maximum temperature of 8.6 °C (47.5 °F), the lowest of March since its record began in 1938 (based on recent data of National Oceanic and Atmospheric Administration, USA). Most parts of the world are experiencing more than above-average or less than below-average temperature in the last two decades based on 140 years of temperature data (Figure 1). For instance, a comparison of 2000 and 2018 maps revealed that the earth is warming up at a dangerous pace but at the same time some areas are hit by unusual cold temperature (Figure 1). These statistics highlight the anomalous frequency of both low and high temperatures in various parts of the world in recent years.

Today, understanding temperature effects on organisms are important to tackling the ongoing challenge we are facing due to global warming. NOAA temperature statistics correlate with the agricultural economic data published by developed countries in recent years. Abiotic stresses cause the loss of hundreds of millions of dollars annually on a global scale [9]. Heat, drought, low temperature, and flooding caused damages of more than USD 1 billion between 1980 and 2004 [3,10]. In 2009, low temperature alone resulted in approximately USD 150 million of crop damage in Japan [11,12]. Additionally in Japan, early and late frost damages fruit and vegetables by approximately JPY 5 to 6 billion per year [11].

Typically, climate change is perceived as an increase in temperature, but in reality, it should be considered as a temperature anomaly where the earth experiences both high and low ends of temperatures. Furthermore, in plants, temperature responses have been well characterized in shoots but less understood for roots [13,14,15,16]. Temperature changes faster in the air than in soil, however, changes in soil temperature persist for a longer time [17]. As a result, soil temperature plays a major role in crop productivity compared to air temperature [17,18]. In terms of climate change, most of the research and review articles until now put emphasis on high-temperature stress on the aerial part of the plants instead of soil temperature and the hidden half of the plant, the root.

Cold stress is a major limiting factor for crop production worldwide, which is broadly categorized into chilling stress (0–15 °C) and freezing stress (<0 °C) [19,20]. Additionally, to combat the cold-induced damage, some plants developed a unique process called cold acclimation where plants can acquire enhanced resistance to freezing stress when they are exposed to nonlethal low temperatures for a few days [21]. Various aspects of cold stress and the underlying mechanisms linked to these processes including transcriptional regulation, calcium signaling, the role of small molecules, and epigenetic regulation have been extensively covered in some recent reviews [22,23,24,25,26,27]. For chilling or low-temperature stress responses in plants, C-repeat-binding factors (CBF)-mediated pathway is the most studied one and is considered as the primary regulatory pathway [28]. Over the last few decades, most articles related to low-temperature response echoed this idea and many of the hypotheses were validated in the model plant *Arabidopsis thaliana* [28]. Although several articles demonstrated the existence of a CBF-independent cold response pathway [29,30,31,32], this pathway is much less explored, and the biological significance is still elusive.

In this review, we focused on only the low-temperature stress response pathway, tried to summarize our current understandings of this pathway on a whole plant level, highlighted the emerging trend in this field, and provided a comparative discussion on CBF-dependent and independent pathways and their possible roles in engineering low-temperature resilient crops for future.

## 2. Major Low Temperature-Responsive Pathway

Plants’ response to low-temperature stress starts from the plasma membrane. The immediate effect on the plasma membrane involves the alteration of fatty acid and lipid-protein interaction [33]. Another major group of regulators involved in the cold perception and relaying the signal to downstream consists of calcium channels, histidine kinase, receptor kinase, and phospholipases [31,34,35,36]. However, compared to these early cold stress perceptive regulators, a group of transcription factors has been identified to relay downstream signals and regulate a series of downstream gene expressions under cold stress. Transcription factors responding to low-temperature stress were first identified by Shinozaki’s and Thomashow’s groups in the early 1990s. This transcription factor family, which was named dehydration-responsive element binding (DREB) and C-repeat binding factor (CBF) by Shinozaki’s and Thomashow’s groups respectively, encodes DREB1A/CBF3, DREB1B/CBF1, and DREB1C/CBF2 transcription factors [37,38]. Since both names are still used, in this review, we will mention both names for each transcription factor to avoid confusion.

Consistent with the prior knowledge about the DREB1/CBFs-mediated cold-induced gene expression, CRISPR-Cas9 generated single, double, and triple mutants of *DREB1/CBFs* demonstrated a decrease in freezing tolerance [39,40]. In this endeavor, two independent groups tried to find out the comparative importance among these three transcription factors for freezing tolerance. Jia et al. (2016) showed that triple mutant is extremely affected during freezing stress and ranked mutants as *cbfs* > *cbf1,3* > *cbf3* for freezing sensitivity [39]. In the same year, Zhao et al. (2016) also presented that triple mutant has most severe phenotype, but they ranked freezing sensitivity as *cbf123* > *cbf2 cbf3* > *cbf1 cbf3* > *cbf2* > *cbf1/cbf3* based on survival rate. The latter mentioned study emphasized the importance on DREB1C/CBF2 and suggested that DREB1C/CBF2 plays a much more important role in freezing tolerance compared to DREB1B/CBF1 and DREB1A/CBF3 [40]. All these results point them as master regulators of cold-inducible gene expression [37,38,39,40].

Upstream activators/inducers of these master regulators were discovered through a series of elegant experiments. For instance, *CALMODULIN BINDING TRANSCRIPTION ACTIVATOR3*/*Arabidopsis thaliana SIGNAL-RESPONSIVE GENE1* (*CAMTA3/AtSR1*) acts as a positive regulator of *DREB1C/CBF2* expression, and *camta1 camta3* double mutant plants are sensitive to freezing stress [41].

Another well-known regulator of *DREB1*/*CBFs* gene expression is *INDUCER OF CBF EXPRESSION1/SCREAM* (*ICE1/SCREAM*), an MYC-like basic helix-loop-helix transcription factor. *ice1*, which is a dominant mutant with a single amino acid substitution at 236 (arginine to histidine R236→H), was first isolated through a screen of a *firefly luciferase (LUC)* reporter gene driven by *CBF3/DREB1A* promoter. In *ice1*, cold-inducible *DREB1A*/*CBF3* gene expression is repressed but the expression of *DREB1B*/*CBF1* or *DREB1C*/*CBF2* is unaltered [42]. *ice1-1* mutant showed increased sensitivity to chilling and freezing response, while *ICE1* overexpressing transgenic plants, *Super-ICE1*, showed an improved survival rate after the freezing treatment [42]. Based on these results, an ICE1-DREB1A/CBF3 was established as a central regulatory pathway for plants’ cold stress response. Later, Kanaoka et al. [43] isolated *scrm-D* mutant, where the majority of the epidermal cells were transformed into guard cells. Interestingly, the mutation was found to be the same missense mutation, like *ice1-1* mutant (R236→H) [43]. They also found that SCRM1/ICE1 and SCRM2/ICE2 make heterodimers with core stomatal transcription factors. Phenotypic observation of the double mutant, *ice1-2 scrm2-1* revealed no stomatal differentiation in the epidermis [43]. These findings raised some questions about the role of *ICE1* as an inducer of *DREB1A/CBF3*.

Recent work from Kidokoro et al. [43] elegantly demonstrated that the *DREB1A/CBF3* repression in *ice1-1* does not depend on the known ICE (R236→H) mutation. Several lines of evidence were provided in support of their findings. They first used the genetic approach where *ice1-1* was crossed onto a line expressing the *ELUC* reporter fused with regions of the promoter of *DREB1A/CBF3* promoter region. This fused line was named 1AR:*ELUC*. In the F2 population, when plants segregated into wild-type (1AR:*ELUC*): heterozygous:*ice1-1/scrm-D* (R236H) as ~1:2:1 ratio, there was a clear induction of *EPF1* (epidermal patterning factor 1), but not *DREB1A/CBF3* expression. Moreover, neither *ICE1* overexpression nor double loss-of-function mutation of *ICE1* and its homolog *SCRM2* altered *DREB1A* expression [44]. All these findings along with the fact that both *EPF1* and *DREB1A/CBF3* are downstream targets of *ICE1*, challenged the idea of ICE1-mediated induction of *DREB1A/CBF3* [43]. To explain the discrepancy, they further explored the possibility of DNA methylation for repression of *DREB1A/CBF3* in the *ice1-1* mutant background. By assessing the 5-methylcytosine (5mC) levels using bisulfide sequencing, they found that the *DREB1A/CBF3* promoter region is hypermethylated in the *ice1-1* mutant. Consistent with these findings, application of 5mC inhibitor, 5-aza-2′-Deoxycytidine (5azaC), recovers *DREB1A/CBF3* expression [44]. The altered methylation of *DREB1A/CBF3* promoter region possibly resulted from the inverted repeat in the reporter gene present in the *ice1-1*mutant, which is unlinked to the R236H mutation [28,44]. These findings confer a severe blow to the well-established and widely accepted ICE1-DREB1A/CBF3 regulatory pathway and warn that this regulatory model should be thoroughly revalidated without any previous assumptions.

## 3. CBF-Dependent Pathway in Crop Plant Engineering

Since the identification of CBF and CBF-dependent pathway components in the model plant *Arabidopsis thaliana*, these genes are favorite targets of researchers for studying cold and chilling stress responses in plants (Figure 2). Over the last three decades, tremendous efforts were put to generate materials, such as knockout mutants and transgenic plants expressing CBF-related genes (Table 1). Genes from various sources were successfully transformed in Arabidopsis and other crops and variable response was observed against cold stress (Table 2). Interestingly, highest number of genes for CBF regulatory pathway was found in Arabidopsis compared with other tested crop plants (Figure 2).

Overexpression of CBF/DREB1A increased the freezing tolerance in non-acclimated Arabidopsis with a tradeoff of dwarf phenotype [51,52,74]. Constitutive overexpression of either *LeCBF1* or *AtCBF3* in transgenic tomato plants did not increase freezing tolerance, but induced dwarf phenotype. Using several elegant experiments, they also demonstrated that tomato has a complete CBF cold response pathway, but the tomato CBF regulon differs from that of Arabidopsis and appears to be considerably smaller and less diverse in function [59]. Overexpression of *OsDREB1A* in rice also showed a similar trend of increased cold tolerance but dwarf phenotype at an optimal temperature [61]. Overexpression of *TaDREB2,TaDREB3* or *TaCBF5L* under stress-responsive promoters increased the frost tolerance in wheat and barley without affecting the growth [65]. Constitutive expression of *HvCBF4* in rice increased the survival rate of plants after the low-temperature stress [67].

Maize is another major crop plant where low-temperature stress regulation has been explored, but so far only ZmDREB1A has been reported from CBF pathway (Figure 2). ZmDREB1A gene was identified first in maize-based on comparative genomics approach relying on the prior information from Arabidopsis [75]. After the initial identification of ZmDREB1A, its interaction with DNA responsive element [76] and expression after the low-temperature stress induction [68] were tested. These studies confirmed its similar role in low-temperature stress response across plant species. In recent years, Han et al. found that RAFFINOSE SYNTHASE, involved in raffinose biosynthesis and chilling stress, is under the transcriptional regulation of ZmDEB1A [77]. Although the maize genome was published more than 10 years ago, the exploration of finding and characterizing low-temperature responsive genes is at the preliminary stage. Efforts are ongoing to find out the low temperature-responsive genes and loci in maize through genomics, high throughput phenotyping, and natural variations [78,79,80,81].

**Table 2 plants-11-00933-t002:** Hormonal biosynthesis, transport, and signaling genes involved in cold-responsive pathway in both CBF-dependent and -independent manner. FT and CA indicate freezing tolerance and cold acclimation, respectively, as the type of treatments.

Gene	Hormone	CBF Pathway	Reference	Type of Treatment
*YUCCA2*	Auxin	Independent	[82]	4 °C
*YUCCA3*	Auxin	Independent	[82]	4 °C
*YUCCA6*	Auxin	Independent	[82]	4 °C
*YUCCA7*	Auxin	Independent	[82]	4 °C
*IAA14*	Auxin	Independent	[83]	4 °C
*PIN2*	Auxin	Independent	[29]	4 °C
*PIN3*	Auxin	Independent	[29]	4 °C
*ETO1*	Ethylene	Dependent	[84]	FT and CA
*ERF4*	Ethylene	Dependent	[85]	FT and CA
*ERF5*	Ethylene	Dependent	[85]	FT and CA
*AHP2*	Cytokinin	Independent	[36]	1 °C and FT
*AHP3*	Cytokinin	Independent	[36]	1 °C and FT
*AHP5*	Cytokinin	Independent	[36]	1 °C and FT
*AHK2*	Cytokinin	Independent	[35]	1 °C, FT, and CA
*AHK3*	Cytokinin	Independent	[35]	1 °C, FT, and CA
*AHK4*	Cytokinin	Independent	[35]	1 °C, FT, and CA
*ARR1*	Cytokinin	Independent	[36]	1 °C and FT
*ARR5*	Cytokinin	Independent	[36]	1 °C and FT
*ARR7*	Cytokinin	Independent	[36]	1 °C and FT
*ARR15*	Cytokinin	Independent	[36]	1 °C and FT
*CRF2*	Cytokinin	Independent	[86]	1 °C
*CRF3*	Cytokinin	Independent	[86]	1 °C
*ABA1*	Abscisic acid	Dependent	[87]	0 °C
*ABA2*	Abscisic acid	Dependent	[87]	0 °C
*ABA4*	Abscisic acid	Dependent	[87]	0 °C
*AAO3*	Abscisic acid	Dependent	[87]	0 °C
*NCED2*	Abscisic acid	Dependent	[87]	0 °C
*NCED3*	Abscisic acid	Dependent	[87]	0 °C
*NCED5*	Abscisic acid	Dependent	[87]	0 °C
*NCED6*	Abscisic acid	Dependent	[87]	0 °C
*NCED9*	Abscisic acid	Dependent	[87]	0 °C
*MYB96*	Abscisic acid	Dependent	[88]	0 °C, FT, and CA
*HOS15*	Abscisic acid	Dependent	[89]	FT
*BIN2*	Brassinosteroid	Dependent	[90]	CA
*BZR1*	Brassinosteroid	Dependent	[90]	CA
*GA1*	Gibberellic acid	Dependent	[91]	CA
*GA2ox*	Gibberellic acid	Dependent	[91]	CA
*DELLA*	Gibberellic acid	Dependent	[91]	CA
*GAI*	Gibberellic acid	Dependent	[91]	CA
*RGA*	Gibberellic acid	Dependent	[91]	CA
*RGL3*	Gibberellic acid	Dependent	[91]	CA
*DAD1*	Jasmonic acid	Dependent	[82]	4 °C
*AOC*	Jasmonic acid	Dependent	[82]	4 °C
*AOS*	Jasmonic acid	Dependent	[82]	4 °C
*OPR*	Jasmonic acid	Dependent	[82]	4 °C
*LOX*	Jasmonic acid	Dependent	[82]	4 °C
*JAR1*	Jasmonic acid	Dependent	[92]	FT and CA
*COI1*	Jasmonic acid	Dependent	[92]	FT and CA
*JAZ1*	Jasmonic acid	Dependent	[92]	FT and CA
*JAZ4*	Jasmonic acid	Dependent	[92]	FT and CA
*ICS1*	Salicylic acid	Dependent	[93]	4 °C
*PAL1*	Salicylic acid	Dependent	[94]	8 °C
*CPR1*	Salicylic acid	Dependent	[95]	5 °C
*SIZ1*	Salicylic acid	Dependent	[96]	FT and CA

These data on CBF-related transgenics and survival rate after cold treatment highlight the importance of this well-studied pathway. At the same time, the observed variability in cold tolerance in crop plants after expressing CBF regulon along with the presence of a limited number of CBF regulatory components in crop plants raises the possibility that CBF pathway might not be the sole pathway for cold stress response in plants. The speculation about CBF-independent pathway in cold response was further strengthened by the discovery of a series of protein trafficking regulators during cold stress from several independent research groups in last 10 years.

## 4. CBF-Independent Cold-Responsive Pathway

As described earlier, for the cold stress response CBF-dependent pathways always took the helm of the research. Hence, newly identified cold-responsive genes were always tested in reference to the CBF regulatory pathway. One of the major groups of genes identified for cold responsiveness is linked to the biosynthesis, transport, and signaling regulators of hormones (Table 2). As most of these genes were hypothesized and experimentally looked at for their CBF-dependence, they were broadly categorized into two groups: genes functioning in CBF-dependent pathway and genes functioning in CBF-independent pathway. Interestingly, except auxin and cytokinin, all other hormonal responses under cold stress were found to be linked to CBF-dependent pathway (Table 2). Since auxin and cytokinin play vital roles during plant growth and development and the genes regulating the response of these two hormones were found to be independent of the CBF regulon for cold stress-induced developmental alterations, indicating that the CBF-independent pathway may also function in parallel to CBF pathway.

One of the major CBF-independent pathways regulating cold stress response is mediated by intracellular auxin homeostasis. Earlier studies on Arabidopsis inflorescence gravity response under cold stress revealed that cold stress transiently inhibits the rootward auxin transport, which can be completely recovered after removing the cold stress [97,98,99]. Cold stress-induced root growth inhibition was attributed to the altered auxin homeostasis at the root meristem, which results from the transient inhibition of the shootward auxin flow [29]. Through several elegant experiments, the authors also demonstrated that cold stress inhibits the trafficking of a subset of intracellular proteins, including PIN2 and PIN3 proteins that play an indispensable role in shootward auxin transport [29]. This work brought new insight into the cold stress response pathway and indicated that auxin and cellular protein trafficking may be the new players in the cold stress response pathway. This idea was substantiated by the study of Hong et al. [32], where they demonstrated that reestablishment of auxin maximum is required at the quiescent center to promote the new columella stem cell daughter cells (CSDCs), which improves the roots’ ability to withstand cold stress. Consistently, it was also shown that exogenous application of IAA helps to reduce selective CSDCs death and facilitates the root growth recovery after chilling stress [32].

The formation of root auxin gradient and auxin maxima solely depends on the transport of auxin, which is tightly regulated by the trafficking of a subset of PIN proteins such as PIN 1, PIN2 [100,101]. Disruption of the functional activity of these proteins either by mutation or chemical inhibitors results in altered gradient formation and maxima [100,101]. It has already been demonstrated that these proteins continuously cycle between the plasma membrane and cytosol using several trafficking pathways. The classic experiment with a general protein trafficking inhibitor Brefeldin A (BFA) provided compelling evidence in support of PIN trafficking [100,101]. Further, it was shown that this continuous trafficking of PINs is important for its function. Cold stress selectively inhibits the PIN2 and PIN3 trafficking resulting in altered auxin gradient, which affects the root development [29]. Interestingly, this inhibition was found to be transient as the removal of cold stress restored the trafficking and the root growth recovery [29]. Further exploration of the mechanism of cold-induced inhibition of protein trafficking revealed that cold stress specifically targets GNOM, a SEC7-containing ARF-GEF (Guanine nucleotide Exchange Factors for ADP Ribosylation Factor) [30]. GNOM contains six characteristic domains (DCB—Dimerization and Cyclophilin Binding domain, HUS—Homology Upstream of SEC7domain, Secretory7—Catalytic domain of GEF, and HDS1–3—Homology Downstream of SEC7 domains 1–3) [102] and among them, SEC7 domain is conserved across the kingdom and regulates GEF catalytic activity in the membrane [101,103].

Partial loss of function trans-heterozygote GNOM mutant *gnom^B4049/emb30^*^−*1*^ (*gnom^B/E^*), where two mutations reside in the SEC7 domain [104], demonstrates a hypersensitive response to cold stress. The mutations outside the SEC7 domain or mutations in GNOM LIKE (GNL) proteins show wild-type-like cold-responsive phenotype [30]. In contrast, the engineered BFA-resistant transgenic GNOM line [101], which contains a point mutation at 696 position (Methionine to Leucine mutation) of SEC7 domain shows strong resistance to cold stress (Figure 3). These plants can grow and flower even when it is grown under continuous cold stress for more than a month [30]. Such an elevated level of resistance of plants to cold has been demonstrated for the first time, which clearly indicates the importance of this pathway in cold stress. The biochemical and expression analyses of the BFA-resistant transgenic line revealed a surprising finding that this point mutation results in overexpression of GNOM both at transcriptional and translational levels [30]. These findings leadto a hypothesis that the increased trafficking activity of GNOM may help in establishing proper auxin gradient even under cold stress, which aids the plants to withstand cold stress and grow.

By narrowing down the cold stress response to SEC7 domain specific activity of GNOM instead of to general cellular protein trafficking, this study presents a possibility that a general regulatory mechanism for cold response may exist in both plant and animal kingdom. In fact, this finding is reminiscent of evidence from yeast. In *S. cerevisiae*, the P-type ATPase Drs2p is a membrane-localized protein and interacts directly and functionally with ARF GEF Gea2p. A single mutant of *drs2Δ* cells is viable but has a cold-sensitive phenotype. Additionally, the double-mutant *drs2Δgea2Δ* strain is even more low-temperature-sensitive. The Gea2^V698G^ mutant fails to interact with Drs2p and consequently becomes temperature- and brefeldin A (BFA)-sensitive [105]. The study from yeast highlighted the importance of GTPase and BFA-sensitive trafficking pathways in regulating cold stress response for growth. In ARF-GEF, the catalytic domain SEC7 plays pivotal roles for GTPase activity, BFA-sensitive response, and temperature sensitivity. Yeast cells of *sec7–4*, containing mutation within SEC7 domain, demonstrate reduced ARF-GEF activity and temperature sensitivity. In contrast, *sec7–1*, containing mutation outside of SEC7 domain, has temperature-insensitive growth and response [106] (Figure 4). Further implicating a universal role of SEC7 domain in temperature response, SEC7 domain-containing GNOM from *Arabidopsis thaliana* can rescue the temperature-sensitive yeast mutant *gea1–19gea2Δ* [103].

## 5. Protein Trafficking in Cold Response: Cell Biologists’ Endeavor

Both high and low-temperature response in plants is mediated by protein trafficking at the cellular level. For instance, at high temperature, elevated auxin response promotes root growth via increased meristematic cell division, where auxin moves faster towards shootward direction through efflux carrier, PIN FORMED2 (PIN2), and maintains the favorable auxin homeostasis to promote root growth at high temperature [107] (Figure 5). The accelerated shootward auxin transport is a cell biological phenomenon as high temperature promotes PIN2 targeting to the plasma membrane instead of vacuolar accumulation [107]. Because PIN2 vacuolar sorting is regulated by SORTING NEXIN1 (SNX1) [108], *snx1* mutants are resistant to high temperature-mediated root development [107]. Interestingly, cold stress or lower than ambient temperature inhibits root growth [29,30,32,109,110], which is the opposite phenotype of high-temperature stress [107] (Figure 5).

Functional activity of the protein trafficking regulator GNOM and the conserved SEC7 domain in cold stress response supports the existence of alternative regulatory pathways for cold stress response. Consistently, several other labs have independently identified proteins that are not transcriptionally regulated through CBF or other temperature responsive transcription factors. For instance, membrane trafficking components RabA4c is upregulated due to exogenous application of glycine betaine, used for various abiotic stress tolerance, and mutant study further confirmed the involvement of RabA4c and glycine betaine for chilling stress-regulated root growth development [111]. Apart from the model plant, the study from crop plants also demonstrated the identification of genes such as *COLD1*, which regulates G-protein signaling, for chilling stress tolerance in rice. Overexpression of *COLD1* induces chilling stress tolerance and downregulation of *COLD1* causes chilling stress-sensitive rice plants. COLD1 is localized in the plasma membrane and endoplasmic reticulum and interacts with G-protein subunit to activate Ca^2+^ channels [31]. Recently, in Arabidopsis, it has been demonstrated that cuticular wax deficient, *cer3–6*, and over-producer, *dewax*, mutants are sensitive and resistant, respectively, at freezing stress [112]. Cuticular wax provides protection against water loss at night when the stomata are closed and during water-limited conditions. Interestingly, cuticular wax is deposited in a polarized fashion in the cell and predicted to use a distinct trafficking pathway for this purpose. Although the trafficking regulator of polarized deposition of cuticular wax is yet to be identified, a series of recent studies tried to understand and narrow down the candidate genes and locus responsible for cuticular wax biosynthesis and trafficking in maize leaf [113,114,115].

Protein trafficking during low-temperature stress is an emerging field. The major challenge to identifying new protein trafficking players during low-temperature stress is finding the interacting partners. In most cases, these protein-protein interactions are transient or not direct. These obstacles will be easily solved with the advancement of techniques, such as single-cell proteomics and proximity labeling. Recent studies from Arabidopsis have shown the promise of proximity labeling to identify the interacting proteins and cellular regulators [116,117], which were difficult to identify previously. In the bigger picture, identified trafficking regulators are well conserved across the kingdom and execute almost similar cellular functions. Understanding the mechanistic regulation of trafficking regulators during cold stress will help us to decipher their role in other abiotic stress responses and crop plant engineering as well.

## 6. Future Perspective

In the age of global warming, aberrant temperatures have challenged our existence. There are major obstacles to meeting the global food crisis. Unfortunately, our molecular-level understanding of temperature sensitivity for plant growth and development is limited to the model plants *Arabidopsis thaliana* and to a certain pathway in ICE-CBF, which needs to be reevaluated for crop plant engineering. Some of the major advancements in the availability of data and techniques shed some light in this regard. For instance, the availability of one thousand plant transcriptomes has opened the door to identify temperature-responsive genes from a large set of plant species that were previously unknown [118]. Even in the case of identified cellular proteins involved in cold stress response, it is generally difficult to figure out their interacting partners as most of them are supposed to interact transiently. The recent proven method of proximity labeling of protein by TurboID will be helpful to identify other cellular response regulators and interacting partners [119]. Moreover, lower sequencing costs and advanced methods will accelerate the process to scrutinize previously well-studied pathways and mechanisms for temperature response as demonstrated in a recent article [44]. To meet the upcoming food crisis during this challenging time of global warming, it is necessary to expand our molecular understanding of temperature sensitivity in model plants and translate it to the crops plants to develop high- and low-temperature-resistant crop varieties of the future.

## Figures and Tables

**Figure 1 plants-11-00933-f001:**
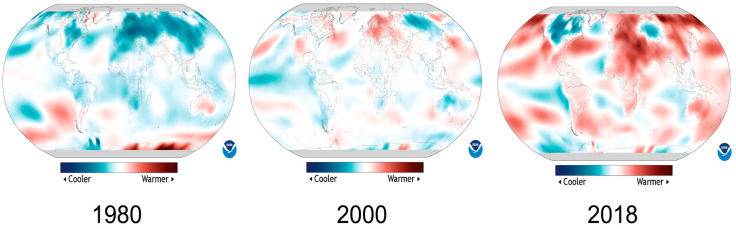
Temperature anomalies around the globe. Colors indicate places where average annual temperature was above or below based on the average temperature during 1981–2010. Data source: Climate.gov (accessed on 28 February 2022). Data provider: NOAA Environmental Visualization Laboratory (NNVL).

**Figure 2 plants-11-00933-f002:**
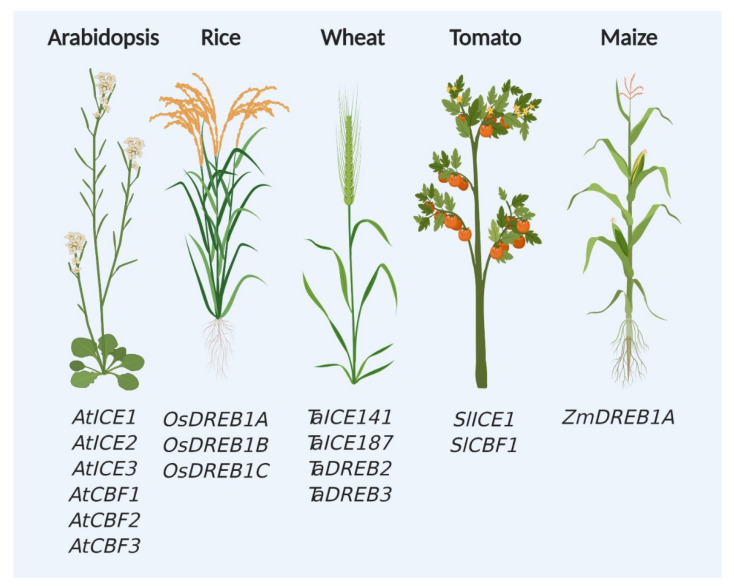
ICE-CBF regulators in the model and crop plants. Identified regulators from the ICE-CBF pathway are highlighted in the model (*Arabidopsis thaliana*) and crop plants (*Oryza sativa, Triticum aestivum, Solanum lycopersicum,* and *Zea mays*).

**Figure 3 plants-11-00933-f003:**
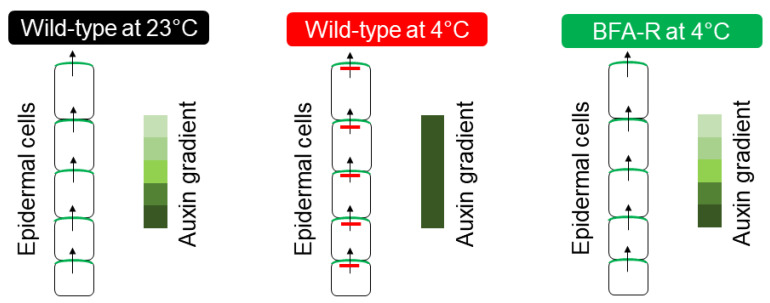
GNOM-mediated faster root growth recovery mechanism under cold stress. (**Left**) Balanced auxin homeostasis in the epidermal cell layers of root at 23 °C. (**Middle**) Low temperature-mediated inhibition of PIN2 trafficking and altered auxin gradient in the epidermal cell layers at the root. (**Right**) GNOM-engineered BFA-resistant line helps to retain functional PIN2 trafficking under cold stress to maintain proper auxin gradient for root growth.

**Figure 4 plants-11-00933-f004:**
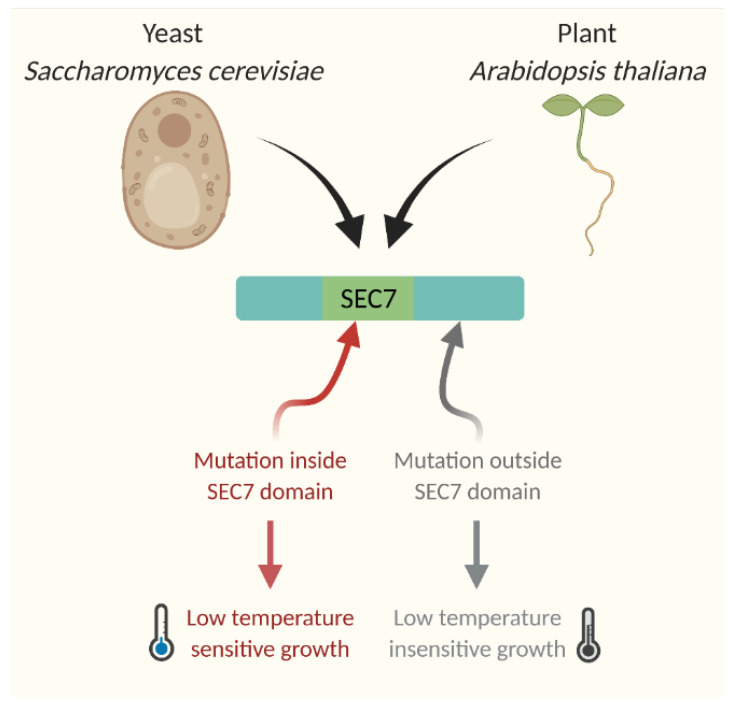
Role of SEC7 domain in temperature response. SEC7 domain-containing proteins are responding to low temperature in both *Saccharomyces cerevisiae* and *Arabidopsis thaliana*.

**Figure 5 plants-11-00933-f005:**
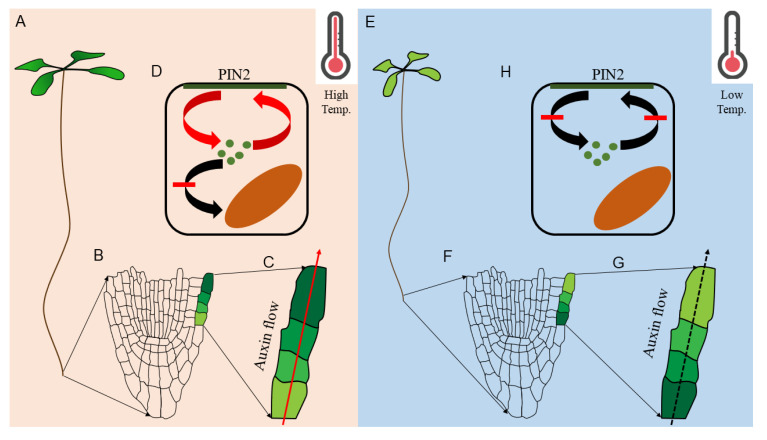
Altered auxin homeostasis in root under high and low temperature. Elongated root growth at high temperature (**A**). Meristematic region and representative epidermal cells (shaded in green color based on auxin level) are demonstrated (**B**) along with the direction of auxin flow and accumulation (**C**). High temperature-mediated PIN2-targeting to the plasma membrane (**D**). Inhibited root growth at low temperature (**E**). Meristematic region and representative epidermal cells (shaded in green color based on auxin level) are demonstrated (**F**) along with the direction of auxin flow and accumulation (**F**). Low temperature mediated PIN2 trafficking inhibition to the plasma membrane (**H**).

**Table 1 plants-11-00933-t001:** ICE and DREB1A/CBF identified from the model plant *Arabidopsis thaliana* and crop plants are validated within the same host plant or other plants for cold tolerance or chilling stress response.

Gene	Source	Host	Reference
*AtICE1*	*Arabidopsis thaliana*	*Arabidopsis thaliana*	[42]
*AtICE2*	*Arabidopsis thaliana*	*Arabidopsis thaliana*	[45]
*AtICE3*	*Arabidopsis thaliana*	*Cucumis sativus*	[46]
*AtDREB1B/* *AtCBF1*	*Arabidopsis thaliana*	*Arabidopsis thaliana*, *Brassica napus*, *Fragaria ananassa*, *Populus tremula x alba*	[47,48,49,50]
*AtDREB1C/* *AtCBF2*	*Arabidopsis thaliana*	*Arabidopsis thaliana*, *Brassica napus*	[48,51]
*AtDREB1A/* *AtCBF3*	*Arabidopsis thaliana*	*Arabidopsis thaliana*, *Brassica napus*, *Solanum tuberosum*, *Triticum aestivum*, *Nicotiana tabacum*, *Manihot esculenta*	[48,52,53,54,55,56]
*SlICE1*	*Solanum lycopersicum*	*Solanum lycopersicum*	[57,58]
*SlCBF1*	*Solanum lycopersicum*	*Arabidopsis thaliana*	[59,60]
*OsDREB1A*	*Oryza sativa*	*Oryza sativa*	[61]
*OsDREB1A*	*Oryza sativa*	*Arabidopsis thaliana*	[62]
*OsDREB1B*	*Oryza sativa*	*Oryza sativa*	[61]
*OsDREB1B*	*Oryza sativa*	*Nicotiana plumbaginifolia*	[63]
*OsDREB1C*	*Oryza sativa*	*Oryza sativa*	[61]
*TaICE141*	*Triticum aestivum*	*Arabidopsis thaliana*	[64]
*TaICE187*	*Triticum aestivum*	*Arabidopsis thaliana*	[64]
*TaDREB2*	*Triticum aestivum*	*Triticum aestivum*	[65]
*TaDREB3*	*Triticum aestivum*	*Triticum aestivum*	[65]
*HvCBF3*	*Hordeum vulgare*	*Arabidopsis thaliana*	[66]
*HvCBF4*	*Hordeum vulgare*	*Oryza sativa*	[67]
*ZmDREB1A*	*Zea mays*	*Arabidopsis thaliana*	[68]
*GmDREB3*	*Glycine max*	*Arabidopsis thaliana*	[69]
*VrCBF1*	*Vitis riparia*	*Arabidopsis thaliana*	[70]
*VrCBF4*	*Vitis riparia*	*Arabidopsis thaliana*	[70]
*LpCBF3*	*Lolium perenne*	*Arabidopsis thaliana*	[71,72]
*MbDREB1*	*Malus baccata*	*Arabidopsis thaliana*	[73]

## Data Availability

Not applicable.

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
