# Peer review of "Cellular Protein Trafficking: A New Player in Low-Temperature Response Pathway"

_plants, 2022, doi:10.3390/plants11070933_

Round 1
Reviewer 1 Report
The review comprises of well-constructed and explained concepts of low temperature response pathways with several examples and summarizing figures.
There are some issues to be dealt with regarding the manuscript. Please see below:
line 329. maize is only once mentioned in the whole MS. What is the reason for it? Could the authors improve the information regarding maize? Are there works presenting other regulators than ZmDREB1A? If so, please include or explain why you would still like to include maize.
Table 2: Colours are unnecessary in the tables. Please remove.
Fig 4: the species should be combined, since the information it contains is the same for Saccharomyces and Arabidopsis
Authors mention proximity labelling in the conclusive part of the MS at the end. Please include a few sentences about it in the main body of the text as well, otherwise it should be omitted.
Author Response
The review comprises of well-constructed and explained concepts of low temperature response pathways with several examples and summarizing figures.
There are some issues to be dealt with regarding the manuscript. Please see below:
line 329. maize is only once mentioned in the whole MS. What is the reason for it? Could the authors improve the information regarding maize? Are there works presenting other regulators than ZmDREB1A? If so, please include or explain why you would still like to include maize.
Response: Thanks for pointing out the issue. In the revised version, we added more information about maize (line 224-236). Hope this will satisfy the reviewer’s concern.
Table 2: Colours are unnecessary in the tables. Please remove.
Response: We removed the colors from both the tables.
Fig 4: the species should be combined, since the information it contains is the same for Saccharomyces and Arabidopsis
Response: Thanks for the suggestion. We combined both the species in the revised figure 4.
Authors mention proximity labelling in the conclusive part of the MS at the end. Please include a few sentences about it in the main body of the text as well, otherwise it should be omitted.
Response: Thanks for pointing out the issue. In the revised version, we added few sentences regarding proximity labeling and single cell proteomics (line 381-388).
Reviewer 2 Report
The manuscript Cellular protein trafficking: a new player in low temperature response pathway by M. Arif Ashraf and Abidur Rahman discusses some aspects of malt stress tolerance. The work represents a relatively narrow aspect of this rather well-studied topic. Unfortunately, the review requires significant proofreading in order to be useful to readers, since the authors did not place restrictions on the aspect under consideration either in the title, or in the introduction or text of the manuscript. Nevertheless, the review can be accepted for consideration and publication with a number of significant additions and a narrowing of the topic.
The most important drawback is the lack of definition of what type or types of cold stress the authors write about. There is a complete lack of understanding of the phenomenon of natural differences in sustainability. I hope that the authors understand that the presence of a certain gene and even its expression does not guarantee the presence of resistance and this should be reflected. The role of osmolytes and calcium signaling was ignored in the work. Well, this is possible, but then it is required to determine the boundaries of the review that would allow such an approach. It is not at all clear why the role of the AFC has not been analyzed.
Also a problem is the lack of consideration of similar responses to completely different abiotic stresses, which allows the use of non-specific resistance enhancement tools to be added.
But the defining shortcoming is ignoring the problems of reaction at the cellular level and protective mechanisms at the cellular level, one gets the feeling that the authors think that all plant cells are the same and do not differ in composition and physical properties. Meanwhile, gene expression will differ in different cells, as will osmotic pressure. In addition, the difference between the effects of low positive temperatures and freezing should be described. otyaby in one paragraph to explain the difference between spring and winter forms, features of cold dormancy. Causes of damage at near-zero and lower temperatures, which are different. Another option is to articulate the topic more clearly and narrow the aspect. In this case, you need to change the name, goals, and conclusions. Then the work can be considered as clearer and more balanced. I hope the authors will choose one of the options for resolving this contradiction that is convenient for them.
Author Response
Reviewer 2
The manuscript Cellular protein trafficking: a new player in low temperature response pathway by M. Arif Ashraf and Abidur Rahman discusses some aspects of malt stress tolerance. The work represents a relatively narrow aspect of this rather well-studied topic. Unfortunately, the review requires significant proofreading in order to be useful to readers, since the authors did not place restrictions on the aspect under consideration either in the title, or in the introduction or text of the manuscript. Nevertheless, the review can be accepted for consideration and publication with a number of significant additions and a narrowing of the topic.
The most important drawback is the lack of definition of what type or types of cold stress the authors write about. There is a complete lack of understanding of the phenomenon of natural differences in sustainability. I hope that the authors understand that the presence of a certain gene and even its expression does not guarantee the presence of resistance and this should be reflected. The role of osmolytes and calcium signaling was ignored in the work. Well, this is possible, but then it is required to determine the boundaries of the review that would allow such an approach. It is not at all clear why the role of the AFC has not been analyzed.
Also a problem is the lack of consideration of similar responses to completely different abiotic stresses, which allows the use of non-specific resistance enhancement tools to be added.
But the defining shortcoming is ignoring the problems of reaction at the cellular level and protective mechanisms at the cellular level, one gets the feeling that the authors think that all plant cells are the same and do not differ in composition and physical properties. Meanwhile, gene expression will differ in different cells, as will osmotic pressure. In addition, the difference between the effects of low positive temperatures and freezing should be described. otyaby in one paragraph to explain the difference between spring and winter forms, features of cold dormancy. Causes of damage at near-zero and lower temperatures, which are different. Another option is to articulate the topic more clearly and narrow the aspect. In this case, you need to change the name, goals, and conclusions. Then the work can be considered as clearer and more balanced. I hope the authors will choose one of the options for resolving this contradiction that is convenient for them.
Response: Thanks for critical reading of the manuscript and comments. We carefully read the reviewer’s comments and tried to accommodate many, if not all of the comments to improve the manuscript.
We agree with the reviewer that there was a lack in defining the cold stress response and mentioning in the beginning that the review is focused on one aspect of cold stress response. To make it clear to the readers, in the revised version, we rewrote the abstract (line 12-34), added the definition of cold stress and clearly mention the scope of this review (line 98-117).
As reviewer mentioned that there are other possible regulatory pathways for the cold stress response, we also are aware of that. However, there are several excellent review articles on those mechanistic pathways. Hence, we did not want to repeat that. In the revised version, we mentioned about it clearly and cited the references accordingly (Line 103-05, ref 22-27).
The other major issues,
[I hope that the authors understand that the presence of a certain gene and even its expression does not guarantee the presence of resistance and this should be reflected.]
Response: We believe that the reviewer misinterpreted our understanding. This is the exact point that we conveyed in this review. In general, researchers think that the sole response pathway for low temperature stress is ICE-CBF pathway and manipulating the genes of this pathway will confer resistance to cold stress irrespective of the plant species. In this review, we presented evidence that this idea is partially correct and there are alternative regulators which should be explored and used to engineer cold stress resistant crop plants.
[But the defining shortcoming is ignoring the problems of reaction at the cellular level and protective mechanisms at the cellular level, one gets the feeling that the authors think that all plant cells are the same and do not differ in composition and physical properties.]
With due respect to the reviewer, we disagree with this comment. In this review, we never mentioned that plant cells are same and do not differ in their composition. What we are trying to convey that for different plants different strategy should be taken and all focuses should not be given on a single regulatory pathway.
[Another option is to articulate the topic more clearly and narrow the aspect. In this case, you need to change the name, goals, and conclusions. Then the work can be considered as clearer and more balanced. I hope the authors will choose one of the options for resolving this contradiction that is convenient for them.]
Thanks for the suggestion. In the revised version, we rewrote the abstract and changed the texts accordingly to narrow down the scope of this review and make it clear to the reader that this review is more focused on low temperature-mediated plant growth response mechanism and present the recent advancement in cell biological studies that has opened a new horizon for finding promising and potential alternative regulators of cold stress response pathway. We believe that the title of the review is clear and should not be changed.
We hope that the revised manuscript will satisfy the reviewer’s concerns.
Round 2
Reviewer 2 Report
In general, the review has become narrower and more specialized. However, I would recommend replacing low temperatures with low positive temperatures, since this is important both from the point of view of physics and from the point of view of biology.
I hope the review will be useful to a number of researchers in this field, and the authors will take into account the aspect of differences between tissues and cells in future work.